# Fermionic neural-network states for ab-initio electronic structure

Kenny Choo[1✉], Antonio Mezzacapo[2✉] & Giuseppe Carleo[3✉]

Neural-network quantum states have been successfully used to study a variety of lattice and continuous-space problems. Despite a great deal of general methodological developments, representing fermionic matter is however still early research activity. Here we present an extension of neural-network quantum states to model interacting fermionic problems. Borrowing techniques from quantum simulation, we directly map fermionic degrees of freedom to spin ones, and then use neural-network quantum states to perform electronic structure calculations. For several diatomic molecules in a minimal basis set, we benchmark our approach against widely used coupled cluster methods, as well as many-body variational states. On some test molecules, we systematically improve upon coupled cluster methods and Jastrow wave functions, reaching chemical accuracy or better. Finally, we discuss routes for future developments and improvements of the methods presented.

[1] Department of Physics, University of Zurich, Winterthurerstrasse 190, 8057 Zurich, Switzerland. [2] IBM Thomas J. Watson Research Center, Yorktown Heights, NY 10598, USA. [3] Center for Computational Quantum Physics, Flatiron Institute, 162 5th Avenue, New York, NY 10010, USA. ✉email: kenny.choo@uzh.ch; amezzac@us.ibm.com; gcarleo@flatironinstitute.org

Predicting the physical and chemical properties of matter from the fundamental principles of quantum mechanics is a central problem in modern electronic structure theory. In the context of ab-initio quantum chemistry (QC), a commonly adopted strategy to solve for the electronic wave-function is to discretize the problem on finite basis functions, expanding the full many-body state in a basis of anti-symmetric Slater determinants. Because of the factorial scaling of the determinant space, exact approaches systematically considering all electronic configurations, such as the full configuration interaction (FCI) method, are typically restricted to small molecules and basis sets. A solution routinely adopted in the field is to consider systematic corrections over mean-field states. For example, in the framework of the coupled cluster (CC) method[1,2], higher level of accuracy can be obtained considering electronic excitations up to doublets, in CCSD, and triplets in CCSD(T). CC techniques are routinely adopted in QC electronic calculations, and they are often considered the "gold standard" in ab-initio electronic structure. Despite this success, the accuracy of CC is intrinsically limited in the presence of strong quantum correlations, in turn restricting the applicability of the method to regimes of relative weak correlations.

For strongly correlated molecules and materials, alternative, non-perturbative approaches have been introduced. Most notably, both stochastic and non-stochastic methods based on variational representations of many-body wave-functions have been developed and constantly improved in the past decades of research. Notable variational classes for QC are Jastrow–Slater wave-functions[3], correlated geminal wave-functions[4], and matrix product states[5–7]. Stochastic projection methods systematically improving upon variational starting points are for example the fixed-node Green's function Monte Carlo[8] and constrained-path auxiliary field Monte Carlo[9]. Main limitations of these methods stem, directly or indirectly, from the choice of the variational form. For example, matrix-product states are extremely efficient in quasi-one-dimensional systems, but suffer from exponential scaling when applied to larger dimensions. On the other hand, variational forms considered so-far for higher dimensional systems typically rely on rigid variational classes and do not provide a systematic and computationally efficient way to increase their expressive power.

To help overcome some of the limitations of existing variational representations, ideas leveraging the power of artificial neural networks (ANN) have recently emerged in the more general context of interacting many-body quantum matter. These approaches are typically based on compact, variational parameterizations of the many-body wave-function in terms of ANN[10]. These approaches to fermionic problems are however comparatively less explored than for lattice spin systems. Two main conceptually different implementations have been put forward. In the first, fermionic symmetry is encoded directly at the mean field level, and ANNs are used as a positive-definite correlator function[11]. Main limitation of this ansatz is that the nodal structure of the wave function is fixed, and the exact ground state cannot, in principle, be achieved, even in the limit of infinitely large ANN. The second method is to use ANNs to indirectly parameterize and modify the fermionic nodal structure[12–15]. In this spirit, "backflow" variational wave functions[16,17] with flexible symmetric orbitals have been introduced[13,14], and only very recently applied to electronic structure[18,19].

In this article, we provide an alternative representation of fermionic many-body quantum systems based on a direct encoding of electronic configurations. This task is achieved by mapping the fermionic problem onto an equivalent spin problem, and then solving the latter with spin-based neural-network quantum states. Using techniques from quantum information, we analyze different model agnostic fermion-to-spin mappings. We show results for several diatomic molecules in minimal Gaussian basis sets, where our approach reaches chemical accuracy (<1 kcal/mol) or better. The current challenges in extending the method to larger basis sets and molecules are also discussed.

## Results

**Electronic structure on spin systems**. We consider many-body molecular fermionic Hamiltonians in second quantization formalism,

$$H = \sum_{i,j} t_{ij}\, c_i^\dagger c_j + \sum_{i,j,k,m} u_{ijkm}\, c_i^\dagger c_k^\dagger c_m c_j, \tag{1}$$

where we have defined fermionic annihilation and creation operators with the anticommutation relation $\{c_i^\dagger, c_j\} = \delta_{i,j}$ on $N$ fermionic modes, and one- and two-body integrals $t_{ij}$ and $u_{ijkm}$. The Hamiltonian in Eq. (1) can be mapped to interacting spin models via the Jordan–Wigner[20] mapping, or the more recent parity or Bravyi–Kitaev[21] encodings, which have been developed in the context of quantum simulations. These three encodings can all be expressed in the compact form

$$c_j \to \tfrac{1}{2} \prod_{i\in U(j)} \sigma_i^x \times \left( \sigma_j^x \prod_{i\in P(j)} \sigma_i^z - i\sigma_j^y \prod_{i\in R(j)} \sigma_i^z \right)$$
$$c_j^\dagger \to \tfrac{1}{2} \prod_{i\in U(j)} \sigma_i^x \times \left( \sigma_j^x \prod_{i\in P(j)} \sigma_i^z + i\sigma_j^y \prod_{i\in R(j)} \sigma_i^z \right), \tag{2}$$

where we have defined an update $U(j)$, parity $P(j)$, and remainder $R(j)$ sets of spins, which depend on the particular mapping considered[22,23], and $\sigma_i^{(x,y,z)}$ denote Pauli matrices acting on site $i$. In the familiar case of the Jordan–Wigner transformation, the update, parity, and remainder sets become $U(j) = j$, $P(j) = \{0, 1, \dots j-1\}$, $R(j) = P(j)$, and the mapping takes the simple form

$$c_j \to \left( \prod_{i=0}^{j-1} \sigma_i^z \right) \sigma_j^-$$
$$c_j^\dagger \to \left( \prod_{i=0}^{j-1} \sigma_i^z \right) \sigma_j^+, \tag{3}$$

where $\sigma_j^{+(-)} = (\sigma_j^x + (-)i\sigma_j^y)/2$. For all the spin encodings considered, the final outcome is a spin Hamiltonian with the general form

$$H_q = \sum_{j=1}^{r} h_j l\sigma_j, \tag{4}$$

defined as a linear combination with real coefficients $h_j$ of $\boldsymbol{\sigma}_j$, $N$-fold tensor products of single-qubit Pauli operators $I$, $\sigma^x$, $\sigma^y$, $\sigma^z$. Additionally, under such mappings, there is a one to one correspondence between spin configuration $\overrightarrow{\sigma}$ and the original particle occupations $\overrightarrow{n}_\sigma$. In the following, we will consider the interacting spin Hamiltonian in Eq. (4) as the starting point for our variational treatment.

**Neural-network quantum states**. Once the mapping is performed, we use neural-network quantum states (NQS) introduced in ref. [10] to parametrize the ground state of the Hamiltonian in Eq. (4). One conceptual interest of NQS is that, because of the flexibility of the underlying non-linear parameterization, they can be adopted to study both equilibrium[24,25] and out-of-equilibrium[26–31] properties of diverse many-body quantum systems. In this work, we adopt a simple neural-network parameterization in terms of a complex-valued, shallow restricted Boltzmann machine (RBM)[10,32]. For a system of $N$ spins, the

many-body amplitudes take the compact form

$$\Psi_M(\vec{\sigma}; \mathcal{W}) = e^{\sum_i a_i \sigma_i^z} \prod_{j=1}^{M} 2\cosh\theta_j(\vec{\sigma}), \text{ where} \quad (5)$$

$$\theta_j(\vec{\sigma}) = b_j + \sum_i^N W_{ij}\sigma_i^z. \quad (6)$$

Here, $\mathcal{W}$ are complex-valued network parameters $\mathcal{W} = \{a, b, W\}$, and the expressivity of the network is determined by the hidden unit density defined by $\alpha = M/N$ where $M$ is the number of hidden units. The simple RBM ansatz can efficiently support volume-law entanglement[33–36], and it has been recently used in several applications[37].

One can then train the ansatz given in Eq. (5) with a variational learning approach known as variational Monte Carlo (VMC), by minimizing the energy expectation value

$$E(\mathcal{W}) = \frac{\langle\Psi_M|H_q|\Psi_M\rangle}{\langle\Psi_M|\Psi_M\rangle}. \quad (7)$$

This expectation value can be evaluated using Monte Carlo sampling using the fact that the energy (and, analogously, any other observable) can be written as

$$E(\mathcal{W}) = \frac{\sum_{\vec{\sigma}} E_{\text{loc}}(\vec{\sigma})|\Psi_M(\vec{\sigma})|^2}{\sum_{\vec{\sigma}}|\Psi_M(\vec{\sigma})|^2}, \quad (8)$$

where we have defined the local energy

$$E_{\text{loc}}(\vec{\sigma}) = \sum_{\vec{\sigma}'} \frac{\Psi_M(\vec{\sigma}')}{\Psi_M^*(\vec{\sigma})} \langle\vec{\sigma}'|H_q|\vec{\sigma}\rangle. \quad (9)$$

Given samples $\mathcal{M}$ drawn from the distribution $\frac{|\Psi_M(\vec{\sigma})|^2}{\sum_{\vec{\sigma}}|\Psi_M(\vec{\sigma})|^2}$, the average over the samples $\hat{E}(\mathcal{W}) = \langle E_{\text{loc}}(\vec{\sigma})\rangle_\mathcal{M}$ gives an unbiased estimator of the energy. Note that the computational cost of evaluating the local energy depends largely on the sparsity of the Hamiltonian $H_q$. In generic QC problems, this cost scales in the worst case with $\mathcal{O}(N^4)$, as compared to the linear scaling in typical condensed matter systems with local interaction.

Sampling from $|\Psi_M(\vec{\sigma})|^2$ is performed using Markov chain Monte Carlo (MCMC), with a Markov chain $\vec{\sigma}_0 \to \vec{\sigma}_1 \to \vec{\sigma}_2 \to \dots$ constructed using the Metropolis–Hastings algorithm[38]. Specifically, at each iteration, a configuration $\vec{\sigma}_{\text{prop}}$ is

proposed and accepted with probability

$$P(\vec{\sigma}_{k+1} = \vec{\sigma}_{\text{prop}}) = \min\left(1, \left|\frac{\Psi_M(\vec{\sigma}_{\text{prop}})}{\Psi_M(\vec{\sigma}_k)}\right|^2\right). \quad (10)$$

The sample $\mathcal{M}$ then corresponds to the configurations of the Markov chain downsampled at an interval $K$, i.e., $\{\vec{\sigma}_0, \vec{\sigma}_K, \vec{\sigma}_{2K}, \dots\}$. For the simulations done in this work, we typically use $K = 10N$ with a sample size of approximately 100,000.

Since the Hamiltonians we are interested in have an underlying particle conservation law, it is helpful to perform this sampling in the particle basis $\vec{n}_\sigma$ rather than the corresponding spin basis $\vec{\sigma}$. The proposed configuration $\vec{\sigma}_{\text{prop}}$ at each iteration, then corresponds to a particle hopping between orbitals. Once a stochastic estimate of the expectation values is available, as well as its derivatives w.r.t. the parameters $\mathcal{W}$, the ansatz can be optimized using the stochastic reconfiguration method[39,40], closely related to the natural-gradient method used in machine learning applications[10,41].

**Computational complexity.** The main computational cost of the procedure arises from the evaluation of the local energy (Eq. (5)) of the samples generated. This gives an overall computational complexity of $\mathcal{O}(N_{\text{var}} \times N_{\text{op}} \times N_{\text{samp}})$ where $N_{\text{var}} = MN + M + N$ is the number of parameters in the network, $N_{\text{op}}$ is the number of Pauli strings in the spin Hamiltonian defined by Eq. (4) and $N_{\text{samp}}$ is sample size.

However, as can be seen in Fig. 2, there are only small number of relevant configurations in the wavefunction, thus each sample $\mathcal{M}$ only contains a few unique configurations. By caching amplitudes $\Psi_M(\vec{\sigma})$ the computational cost can be significantly reduced to $\mathcal{O}(N_{\text{var}} \times N_{\text{op}} \times N_{\text{unique}})$ where $N_{\text{unique}} \ll N_{\text{samp}}$ is the average number of unique configurations in each sample. Typically, for a sample size of 10,000 there are only about few hundred unique samples.

**Potential energy surfaces.** We first consider small molecules in a minimal basis set (STO-3G). We show in Fig. 1 the dissociation curves for $C_2$ and $N_2$, compared to the CCSD and CCSD(T). It can be seen that on these small molecules in their minimal basis, the RBM is able to generate accurate representations of the ground states, and remarkably achieve an accuracy better than standard QC methods. To further illustrate the expressiveness of the RBM, we show in Fig. 2 the probability distribution of the most relevant configurations in the wavefunction. We contrast between the RBM and configuration interaction limited to single

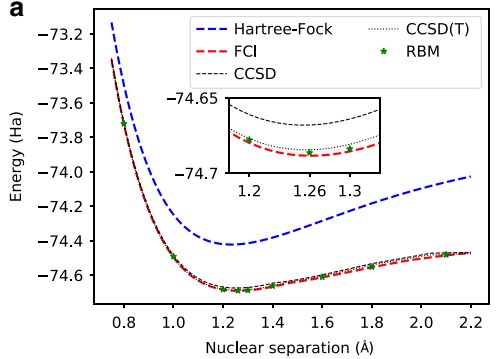
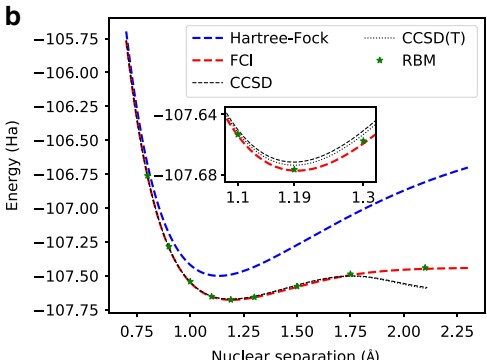

**Fig. 1 Dissociation profiles.** The accuracy of fermionic neural-network quantum states compared with other quantum chemistry approaches. Shown here are dissociation curves for **a** $C_2$ and **b** $N_2$, in the STO-3G basis with 20 spin-orbitals. The RBM used has 40 hidden units, and it is compared to both coupled-cluster approaches (CCSD, CCSD(T)) and FCI energies.

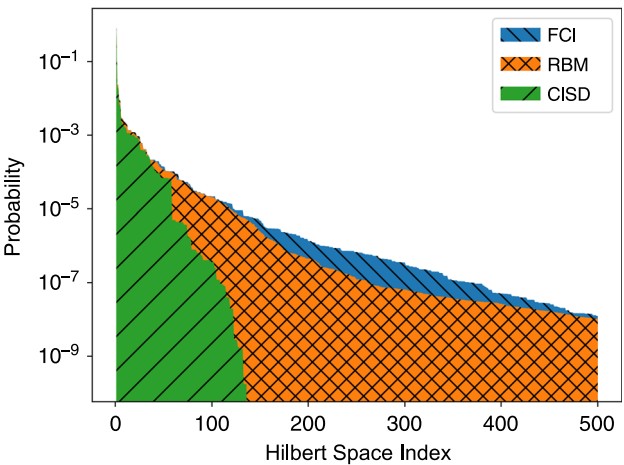

**Fig. 2 Electronic correlations.** Probabilities (in logarithmic scale) of the 500 most probable configurations in the FCI (blue), RBM (orange), and CISD (green) wavefunctions for the equilibrium nitrogen $N_2$ molecule in the STO-3G basis.

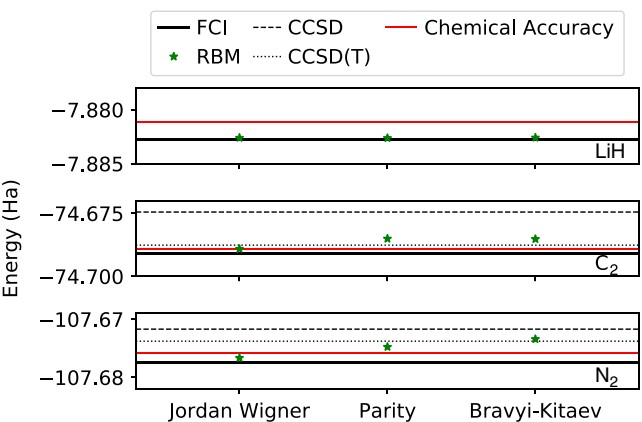

**Fig. 3 Comparison of different spin mappings.** Accuracy of the RBM (green star) representations for three different mapping types (Jordan–Wigner, Parity, and Bravyi–Kitaev) and three different molecules (LiH, $C_2$, and $N_2$) in their equilibrium configuration in the STO-3G basis. The geometries used are reported in the Methods section.

and double excitations (CISD). In CISD, the Hilbert space is truncated to include only states which are up to two excitations away from the Hartree–Fock configuration. It is clear from the histogram that the RBM is able to capture correlations beyond double excitations.

**Alternative encodings**. The above computations were done using the Jordan–Wigner mapping. To investigate the effect of the mapping choice on the performance of the RBM, we also performed select calculations using the parity and Bravyi–Kitaev mappings. All the aforementioned transformations require a number of spins equal to the number of fermionic modes in the model. However, the support of the Pauli operators $w_j = |\sigma_j|$ in Eq. (4), i.e., the number of single-qubit Pauli operators in $\sigma_j$ that are different from the identity $I$, depends on the specific mapping used. Jordan–Wigner and parity mappings have linear scalings $w_j = O(N)$, while the Bravyi–Kitaev encoding has a more favorable scaling $w_j = O(\log(N))$, due to the logarithmic spin support of the update, parity, and remainder sets in Eq. (2). Note that one could in principle use generalized superfast mappings[42], which have a support scaling as good as $w_j = O(\log(d))$, where $d$ is the maximum degree of the fermionic interaction graph defined by Eq. (1). However, such a mapping is not practical for the models considered here because the typical large degree of molecular interactions graphs makes the number of spins required for the simulation too large compared to the other model-agnostic mappings.

While these encodings are routinely used as tools to study fermionic problems on quantum hardware[43], their use in classical computing has not been systematically explored so far. Since they yield different structured many-body wave functions, it is then worth analyzing whether more local mappings can be beneficial for specific NQS representations. In Fig. 3, we analyze the effect of the different encodings on the accuracy of the variational ground-state energy for a few representative diatomic molecules. At fixed computational resources and network expressivity, we typically find that the RBM ansatz can achieve consistent levels of accuracy, independent of the nature of the mapping type. While the Jordan–Wigner allows to achieve the lowest energies in those examples, the RBM is nonetheless able to efficiently learn the ground state also in other representations, and chemical accuracy is achieved in all cases reported in Fig. 3.

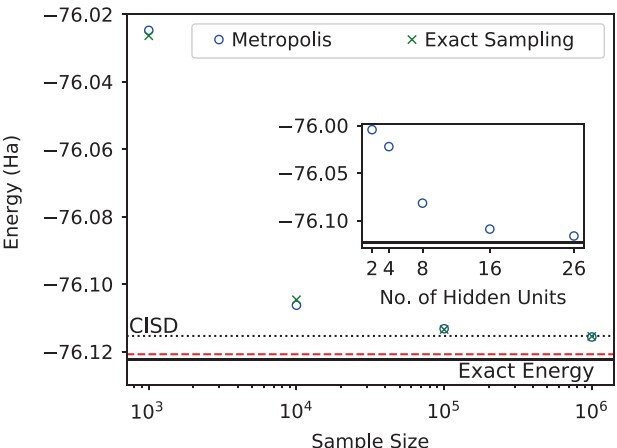

**Fig. 4 Sampling size dependence of the converged energies.** Converged energy of $H_2O$ in the 6-31g basis (26 spin-orbitals) as the number of samples used for each VMC iteration is varied. The converged energy for the samples obtained using the Metropolis algorithm (blue circles) matches that obtained using exact sampling (green crosses), beating the accuracy of CISD and approaching chemical accuracy (red line) for the largest sample size. In the inset, we also show the variational energy as the number of hidden units is increased from 2 to 26.

**Sampling larger basis sets**. The spin-based simulations of the QC problems studied here show a distinctive MCMC sampling behavior that is not usually found in lattice model simulations of pure spin models. Specifically, the ground-state wave function of the diatomic molecules considered is typically sharply peaked around the Hartree–Fock state, and neighboring excited states. This behavior is prominently shown also in Fig. 2, where the largest peaks are several order of magnitude larger than the distribution tail. As a result of this structure, any uniform sampling scheme drawing states $\vec{\sigma}$ from the VMC distribution $|\Psi_M(\vec{\sigma})|^2$, is bound to repeatedly draw the most dominant states, while only rarely sampling less likely configurations. To exemplify this peculiarity, we study the behavior of the ground state energy as a function of the number of MCMC samples used at each step of the VMC optimization. We concentrate on the water molecule in the larger 6-31g basis. In this case, the Metropolis sampling scheme exhibits acceptance rates as low as 0.1% or less, as a

**Table 1 Equilibrium energies (in Hartree) as obtained by different methods.**

| Molecule | RBM | Jastrow | CISD | CCSD | CCSD(T) | FCI |
|---|---|---|---|---|---|---|
| $H_2$ | −1.1373 | −1.1373 | −1.1373 | −1.1373 | −1.1373 | −1.1373 |
| LiH | −7.8826 | −7.8814 | −7.8827 | −7.8828 | −7.8828 | −7.8828 |
| $NH_3$ | −55.5277 | −55.4770 | −55.5258 | −55.5280 | −55.5281 | −55.5282 |
| $H_2O$ | −75.0232 | −74.9784 | −75.0221 | −75.0231 | −75.0232 | −75.0233 |
| $C_2$ | −74.6892 | −74.5001 | −74.6371 | −74.6745 | −74.6876 | −74.6908 |
| $N_2$ | −107.6767 | −107.5924 | −107.6591 | −107.6717 | −107.6738 | −107.6774 |

The basis set considered here is STO-3G, and the corresponding geometries are reported in the Methods section. Energies are reported in Hartrees and statistical uncertainty on RBM and Jastrow states energies are on the last reported digits. The RBM used has a hidden unit density $\alpha = 1$ for all the molecules apart from $C_2$ and $N_2$ where we use $\alpha = 2$.

**Table 2 Equilibrium configurations used for the ground-state calculations presented in the main text. The coordinates (x, y, z) are given in angstroms (Å).**

| Molecule | Basis | Geometry |
|---|---|---|
| $H_2$ | STO-3G | H(0, 0, 0) |
| | | H(0, 0, 0.734) |
| LiH | STO-3G | Li(0, 0, 0) |
| | | H(0, 0, 1.548) |
| $NH_3$ | STO-3G | N(0, 0, 0.149) |
| | | H(0, 0.947, −0.348) |
| | | H(0.821, −0.474, −0.348) |
| | | H(−0.821, −0.474, −0.348) |
| $C_2$ | STO-3G | C(0, 0, 0) |
| | | C(0, 0, 1.26) |
| $N_2$ | STO-3G | N(0, 0, 0) |
| | . | N(0, 0, 1.19) |
| | | H(0, 0.769, −0.546) |
| $H_2O$ | STO-3G | H(0, −0.769, −0.546) |
| | | O(0, 0, 0.137) |
| | | H(0, 0.795, −0.454) |
| $H_2O$ | 6-31G | H(0, −0.795, −0.454) |
| | | O(0, 0, 0.113) |

consequence of the presence of dominating states previously discussed.

In Fig. 4, we vary the sample size and also compare MCMC sampling with exact sampling. We can see that the accuracy of the simulation depends quite significantly on the sample size. The large number of samples needed in this case, together with a very low acceptance probability for the Metropolis–Hasting algorithm, directly points to the inefficiency of uniform sampling from $|\Psi_M(\vec{\sigma})|^2$. At present, this represents the most significant bottleneck in the application of our approach to larger molecules and basis sets. This issue however is not a fundamental limitation, and alternatives to the standard VMC uniform sampling can be envisioned to efficiently sample less likely—yet important for chemical accuracy—states. Beyond sampling issues, representability is also a factor as can be seen from the inset of Fig. 4. Enough hidden units are required to capture the wavefunction accurately, however, with more hidden units optimization also becomes more challenging, thus finding an appropriate network architecture is also crucial.

## Discussion

In this work, we have shown that relatively simple shallow neural networks can be used to compactly encode, with high precision, the electronic wave function of model molecular problems in quantum chemistry. Our approach is based on the mapping between the fermionic quantum chemistry molecular Hamiltonian and corresponding spin Hamiltonians. In turn, the ground state of the spin models can be conveniently modeled with standard variational neural-network quantum states. On model diatomic molecules, we show that a RBM state is able to capture almost the entirety of the electronic excitations, improving on routinely used approaches as CCSD(T) and the Jastrow ansatz (Table 1).

Several future directions can be envisioned. The distinctive peaked structure of the molecular wave function calls for the development of alternatives to uniform sampling from the Born probability. These developments will allow to efficiently handle larger basis sets than the ones considered here. Second, our study has explored only a very limited subset of possible neural-network architectures. Most notably, the use of deeper networks might prove beneficial for complex molecular complexes. Another very interesting matter for future research is the comparison of different neural-network-based approaches to quantum chemistry. Contemporary to this work, approaches based on antisymmetric wave-functions in continuous space have been presented[18,19]. These have the advantage that they already feature a full basis set limit. However, the discrete basis approach has the advantage that boundary conditions and fermionic symmetry are much more easily enforced. As a consequence, simple-minded shallow networks can already achieve comparatively higher accuracy than the deeper and substantially more complex networks so-far adopted in the continuum case. On a different note, in a recent article[44], the use of a unitary-coupled RBM applicable for noisy intermediate-scale quantum devices has been proposed and is also worth exploring.

## Methods

**Geometries for diatomic molecules**. The equilibrium geometries for the molecules presented in this work were obtained from the CCCBDB database [45]. For convenience, we present them in Table 2.

**Computing matrix elements**. A crucial requirement for the efficient implementation of the stochastic variational Monte Carlo procedure to minimize the ground-state energy, is the ability to efficiently compute the matrix elements of the spin Hamiltonian $\langle \vec{\sigma}' | H_q | \vec{\sigma} \rangle$, appearing in the local energy, Eq. (9). Since $H_q$ is a sum of products of Pauli operators, the goal is to efficiently compute matrix elements of the form

$$\mathcal{M}(\vec{\sigma}, \vec{\sigma}') = \langle \vec{\sigma}' | \sigma_1^{\nu_1} \sigma_2^{\nu_2} \dots \sigma_N^{\nu_N} | \vec{\sigma} \rangle, \tag{11}$$

where $\sigma_i^{\nu_i}$ denotes a Pauli matrix with $\nu = I, x, y, z$ acting on site $i$. Because of the structure of the Pauli operators, these matrix elements are non-zero only for a specific $\vec{\sigma}'$ such that

$$\begin{cases} \sigma_i' = \sigma_i & \nu_i \in (I, Z) \\ \sigma_i' = -\sigma_i & \nu_i \in (X, Y) \end{cases} \tag{12}$$

and the matrix element is readily computed as

$$\mathcal{M}(\vec{\sigma}, \vec{\sigma}') = (i^{n_y}) \prod_{k:\nu_k \in (y,z)} \sigma_k, \tag{13}$$

where $n_y$ is the total number of $\sigma^y$ operators in the string of Pauli matrices.

**Simulation details**. The optimization follows the stochastic reconfiguration scheme as detailed in the supplementary material of ref. [10]. Given a variational ansatz $\Psi(\{\alpha_k\}) \in \mathbb{C}^{2^n}$ depending on parameters $\{\alpha_k\}$, the parameter update $\delta\alpha_k$ is given by solution of the linear equation

$$\sum_{k'} \left[ \langle \mathcal{O}_k^\dagger \mathcal{O}_{k'} \rangle - \langle \mathcal{O}_k^\dagger \rangle \langle \mathcal{O}_{k'} \rangle + \lambda \delta_{kk'} \right] \delta\alpha_{k'}$$
$$= -\epsilon \left[ \langle \mathcal{O}_k^\dagger \hat{H} \rangle - \langle \mathcal{O}_k^\dagger \rangle \langle \hat{H} \rangle \right], \quad (14)$$

where $\mathcal{O}_k = \frac{\partial}{\partial \alpha_k} \log \left[ \Psi(\{\alpha_k^0\}) \right]$ are the logarithmic derivatives, $\epsilon$ is the step size and $\lambda$ is the regularization parameter. For the simulations done in this paper, we take $\epsilon = 0.05$ and $\lambda = 0.01$. The expectation values $\langle \cdots \rangle$ are estimated with Markov chain Monte Carlo sampling as described in the main text.

The parameters of the RBM are initialized from a random normal distribution with a zero mean and a standard deviation of 0.05.

## Data availability
The datasets generated during and/or analyzed during the current study are available from the authors on reasonable request.

## Code availability
The code used in the current study is largely based on the open-sourced software NetKet[46] with some custom modifications, which will be made available from the authors upon reasonable request.

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

## Acknowledgements
The Flatiron Institute is supported by the Simons Foundation. A.M. acknowledges support from the IBM Research Frontiers Institute. K.C. is supported by the European Unions' Horizon 2020 research and innovation program (ERC-StG-Neupert-757867-PARATOP). Neural-network quantum states simulations are based on the open-source software NetKet[46]. Coupled cluster and configuration interaction calculations are performed using the PySCF package[47]. The mappings from fermions to spins are done using Qiskit Aqua[48]. The authors acknowledge discussions with G. Booth, T. Berkelbach, M. Holtzmann, J. E. T. Smith, S. Sorella, J. Stokes, and S. Zhang.

## Author contributions

K.C. performed the numerical simulations. K.C., A.M., and G.C. devised the algorithm and wrote the manuscript.

## Competing interests

The authors declare no competing interests.
