## [Peer Review File · Nature Communications]

Reviewers' comments:

Reviewer #1 (Remarks to the Author):

The authors provide a systematic way to use the restricted Boltzmann machine (RBM) to obtain accurate electronic structure calculations of Fermionic systems. This is done by modifying the energy function of RBM to make it a complex-valued function. They also achieved high accuracy in calculating the ground state energies of different small molecules. This paper also compared how different encoding transformation as well as a number of samplings would affect the final results.

One important issue that the author should address is the complexity of the current RBM approach compared to the traditional ab initio approaches used in this study such as CCSD and FCI. The sampling is enabled by Metropolis-Hasting algorithm but MCMC algorithm may need a large sample size to achieve high accuracy. As Fig4 shows, the sample size needs to be 10^6 to achieve error below the chemical accuracy.

I'd like to draw the authors attention to a new paper on the arXiv reporting a modification of RBM: Unitary-Coupled Restricted Boltzmann Machine Ansatz for Quantum Simulations, arXiv:1912.02988.

The paper is well written with an interesting new approach and accurate results and should be published after the authors address the complexity issue discussed above.

Reviewer #2 (Remarks to the Author):

I have very much enjoyed reading the paper "Fermionic neural-network states for ab-initio electronic structure " by Choo, Mezzacapo and Carleo. The paper considers a second-quantization approach to molecular electronic structure calculations and represents the arising spin Hamiltonian with a Restricted Boltzmann Machine (RBM) that is trained with variational Monte Carlo, following the approach by Carleo and Troyer (Science 2017) that was previously only used for lattice systems.

The method is a very original and interesting contribution and nice complement to the recent deep QMC approaches to electronic structure by Hermann et al and Pfau et al. The paper is insightful, well written and very honest about the merits and limitations of the approach. I particularly like Figure 2 that shows how the RBM approach more efficiently samples the long tail of spin states needed to reach very high accuracy in the variational energy that is missed by CISD.

I propose to accept this paper to Nature Communications subject to a few changes:

1. Minimal Basis Set 1: Most results are for a very small basis set. Within this basis set, the RBM approach is shown to reach FCI accuracy and the correlation energy is recovered, but while discussing these results, it should be noted, that these energies are very different from the energies achieved with a large basis set. For example, in the large basis limit, FCI results for H₂ and LiH are -1.1744 [Kolos and Wolniewicz, JCP 43:2429(1965)] and -8.0705 [Casalegno et al, JCP 118:7193(2003)] compared to the -1.1373 and -7.8828 reported here. Also the terms "exact energy" and "correlation energy" are used here with respect to a fixed basis set, I think these terms should be used a bit more careful, because it is IMO confusing to use them with a small basis set and to a non-expert reading this will suggest that the authors are approaching the exact solution of the electronic Schrödinger equation rather than a fixed-basis set approximation of it.
2. Minimal Basis Set 2 (optional): While the paper discusses sampling issues arising in larger basis sets for the H₂O molecule, it would be interesting to see if the current approach can recover nearly exact results for these small molecules. While the "sampling larger basis sets" section indicates

that the current sampling problems can be addressed, the paper would benefit a lot from a demonstration that the current method gives competitive high-accuracy results at least for the small systems. I don't want to set this as a hard requirement for acceptance though.

3. Technical details: I believe the paper is missing the technical details required to reproduce the results. I suggest that the authors include a short appendix describing the main choices such as neural network structure and hyperparameters. As the main software to reproduce the results is available (NetKet), the authors should publish the scripts or notebooks used to produce the current results with NetKet.

Frank Noe

Reviewer #3 (Remarks to the Author):

In this manuscript, Carleo and coworkers extend the recently developed method of using RBMs to model quantum many-body states to systems of fermions. The key advance in this work is to use a fermion-to-spin mapping, and then solve the non-local spin hamiltonian with the RBM. The authors show that this approach is able to describe electron correlation with quite respectable accuracy, with higher accuracy results presumably possible using more hidden nodes. This manuscript is very nicely written and provides important and timely results. I recommend publication after addressing the following minor comments:

1) "We show results for several diatomic molecules in minimal Gaussian basis sets, where our approach reaches chemical accuracy (< 5 kcal/mol) or better."

The common definition for "chemical accuracy" is 1 kcal/mol, not 5. While this is somewhat arbitrary, 1 kcal/mol is more appropriate conventionally.

2) I assume that $\alpha=1$ was chosen due to computational complexity. However, because the accuracy of the results depends on the depth of the hidden layer, it would be helpful to see some discussion of the computational cost associated with increasing the hidden layer density.

3) It might be helpful to the reader to include the CISD energies in Table I, so that the energetic consequences of Fig 2, can be seen.

4) I think the section titled "Alternative encodings" is particularly interesting, yet could use a bit more discussion. In this section I had a couple questions:

a) My understanding (which could be lacking), is that the only reason one would choose the BK mapping is for computational considerations. Did the authors notice any computational speedup from using BK?

b) Is there an ordering dependence on the RBM? For instance, if you re-order the fermionic modes do you get the exact same result? Assuming each visible node connects to each hidden node, I could imagine this being ordering-invariant, but I can't really tell. If it's not, the authors should try to evaluate the accuracy sensitivity to fermionic mode ordering.

c) How does the chosen mapping impact the particle number preservation in the sampling? The statement "it is helpful to perform this sampling in the particle basis $\vec{n}\sigma$ rather than the corresponding spin basis $\vec{\sigma}$." describes a simple way to enforce particle number symmetry. However, this symmetry is only local in the JW transformation. Was this also done for the BK or parity mappings? If not, could this impact the results detailed in Fig 3?

Nick Mayhall

Reviewer #1 (Remarks to the Author):

The authors provide a systematic way to use the restricted Boltzmann machine (RBM) to obtain accurate electronic structure calculations of Fermionic systems. This is done by modifying the energy function of RBM to make it a complex-valued function. They also achieved high accuracy in calculating the ground state energies of different small molecules. This paper also compared how different encoding transformation as well as a number of samplings would affect the final results.

One important issue that the author should address is the complexity of the current RBM approach compared to the traditional ab initio approaches used in this study such as CCSD and FCI. The sampling is enabled by Metropolis-Hasting algorithm but MCMC algorithm may need a large sample size to achieve high accuracy. As Fig4 shows, the sample size needs to be 10^6 to achieve error below the chemical accuracy.

While it is true the MCMC algorithm may requires a large sample size to achieve high accuracy, there are much fewer unique configurations within the sample. By making use of caches, the computational cost can be greatly reduced. We have added some statements regarding the computational complexity of the algorithm.

I'd like to draw the authors attention to a new paper on the arXiv reporting a modification of RBM: Unitary-Coupled Restricted Boltzmann Machine Ansatz for Quantum Simulations, arXiv:1912.02988.

We thank the referee for bringing this relevant paper to our attention. We have added a citation to that paper.

The paper is well written with an interesting new approach and accurate results and should be published after the authors address the complexity issue discussed above.

We are grateful for the positive review and feedback on our work, we hope that the revised version is now in good shape to be published.

Reviewer #2 (Remarks to the Author):

I have very much enjoyed reading the paper "Fermionic neural-network states for ab-initio electronic structure" by Choo, Mezzacapo and Carleo. The paper considers a second-quantization approach to molecular electronic structure calculations and represents the arising spin Hamiltonian with a Restricted Boltzmann Machine (RBM) that is trained with variational Monte Carlo, following the approach by Carleo and Troyer (Science 2017) that was previously only used for lattice systems.

The method is a very original and interesting contribution and nice complement to the recent deep QMC approaches to electronic structure by Hermann et al and Pfau et al. The paper is insightful, well written and very honest about the merits and limitations of the approach. I particularly like Figure 2 that shows how the RBM approach more efficiently samples the long tail of spin states needed to reach very high accuracy in the variational energy that is missed by CISD.

I propose to accept this paper to Nature Communications subject to a few changes:

We sincerely thank Frank Noé for the encouraging statements and for the positive review and valuable feedback. We address in the following his remarks and highlight the changes to the text.

1. Minimal Basis Set 1: Most results are for a very small basis set. Within this basis set, the RBM approach is shown to reach FCI accuracy and the correlation energy is recovered, but while discussing these results, it should be noted, that these energies are very different from the energies achieved with a large basis set. For example, in the large basis limit, FCI results for H₂ and LiH are -1.1744 [Kolos and Wolniewicz, JCP 43:2429(1965)] and -8.0705 [Casalegno et al, JCP 118:7193(2003)] compared to the -1.1373 and -7.8828 reported here. Also the terms "exact energy" and "correlation energy" are used here with respect to a fixed basis set, I think these terms should be used a bit more careful, because it is IMO confusing to use them with a small basis set and to a non-expert reading this will suggest that the authors are approaching the exact solution of the electronic Schrödinger equation rather than a fixed-basis set approximation of it.

We have removed the use of the terms "exact energy" and "correlation energy" to avoid any potential source of confusion. Hopefully it is now clearer for the reader that our accuracies statements are referring to the fixed basis sets.

2. Minimal Basis Set 2 (optional): While the paper discusses sampling issues arising in larger basis sets for the H₂O molecule, it would be interesting to see if the current approach can recover nearly exact results for these small molecules. While the "sampling larger basis sets" section indicates that the current sampling problems can be addressed, the paper would benefit a lot from a demonstration that the current method gives competitive high-accuracy results at least for the small systems. I don't want to set this as a hard requirement for acceptance though.

3. Technical details: I believe the paper is missing the technical details required to reproduce the results. I suggest that the authors include a short appendix describing the main choices such as neural network structure and hyperparameters. As the main software to reproduce the results is available (NetKet), the authors should publish the scripts or notebooks used to produce the current results with NetKet.

We have followed the suggestion and now added a more detailed appendix on the technical details of the method. While the work is based on NetKet, we are using here a highly customized version of the code that is currently not compatible with the upstream master version of the code. We are in the process of preparing a customized version which will be ready soon, but we leave (a proper) integration with NetKet for later work.

Frank Noe

Reviewer #3 (Remarks to the Author):

In this manuscript, Carleo and coworkers extend the recently developed method of using RBMs to model quantum many-body states to systems of fermions. The key advance in this work is to use a fermion-to-spin mapping, and then solve the non-local spin hamiltonian with the RBM. The authors show that this approach is able to describe electron correlation with quite respectable accuracy, with higher accuracy results presumably possible using more hidden nodes. This manuscript is very nicely written and provides important and timely results. I recommend publication after addressing the following minor comments:

We sincerely thank Nick Mayhall for the positive evaluation of our work and for the valuable feedback.

1) "We show results for several diatomic molecules in minimal Gaussian basis sets, where our approach reaches chemical accuracy (< 5 kcal/mol) or better."
The common definition for "chemical accuracy" is 1 kcal/mol, not 5. While this is somewhat arbitrary, 1 kcal/mol is more appropriate conventionally.

2) I assume that $\alpha=1$ was chosen due to computational complexity. However, because the accuracy of the results depends on the depth of the hidden layer, it would be helpful to see some discussion of the computational cost associated with increasing the hidden layer density.

Indeed $\alpha=1$ was chosen for computational reasons. We have added some comments in the text regarding the computational complexity of increasing the number of hidden units as well as the overall computational complexity of the algorithm.

3) It might be helpful to the reader to include the CISD energies in Table I, so that the energetic consequences of Fig 2, can be seen.

We have added a column for the CISD energies in Table 1.

4) I think the section titled "Alternative encodings" is particularly interesting, yet could use a bit more discussion. In this section I had a couple questions:

a) My understanding (which could be lacking), is that the only reason one would choose the BK mapping is for computational considerations. Did the authors notice any computational speedup from using BK?

Indeed, the support of the operators from the BK mapping tend to be smaller than that of Jordan-Wigner or parity. However the computational costs is mainly determined by the number of operators rather than the support of the operators, as such, there wasn't any noticeable computational speed up.

b) Is there an ordering dependence on the RBM? For instance, if you re-order the fermionic modes do you get the exact same result? Assuming each visible node connects to each hidden node, I could imagine this being ordering-invariant, but I can't really tell. If it's not, the authors should try to evaluate the accuracy sensitivity to fermionic mode ordering.

The effect of reordering the fermionic modes is somewhat similar to the effect of using a different fermion-to-spin mapping. The result of which can be seen in Fig. 3. For the mappings tested, the results seem to suggest that the RBM is relatively agnostic towards the mapping. However, we tend to find more accurate results within the Jordan Wigner mapping framework. At this stage we do not have an intuition as for why this is the case, but we hope that this observation will stimulate further research both in the quantum chemistry and in the quantum computing community.

c) How does the chosen mapping impact the particle number preservation in the sampling? The statement "it is helpful to perform this sampling in the particle basis \vec{n} rather than the corresponding spin basis $\vec{\sigma}$." describes a simple way to enforce particle number symmetry. However, this symmetry is only local in the JW transformation. Was this also done for the BK or parity mappings? If not, could this impact the results detailed in Fig 3?

The mappings we used in the paper are one-to-one, such that each particle configuration corresponds to a unique spin configuration and vice versa. In this way, regardless of the mapping used, (JW, BK or Parity) we can always map the spin configuration back to the particle configuration, perform a particle number conserving operation and then map it back to the spin configuration. We have also added an appendix containing more technical details on the algorithms used.

Nick Mayhall

REVIEWERS' COMMENTS:

Reviewer #1 (Remarks to the Author):

They have addressed all comments and should be published

Sabre Kais

Reviewer #2 (Remarks to the Author):

The manuscript is acceptable for publication. Congratulations on the great work.

Frank Noe

Reviewer #3 (Remarks to the Author):